DOI: 10.1038/s41467-018-05225-1　　**OPEN**

# Hollow-core conjoined-tube negative-curvature fibre with ultralow loss

Shou-fei Gao [1], Ying-ying Wang [1], Wei Ding [2], Dong-liang Jiang[1], Shuai Gu[1], Xin Zhang[1] & Pu Wang[1]

Countering the optical network 'capacity crunch' calls for a radical development in optical fibres that could simultaneously minimize nonlinearity penalties, chromatic dispersion and maximize signal launch power. Hollow-core fibres (HCF) can break the nonlinear Shannon limit of solid-core fibre and fulfil all above requirements, but its optical performance need to be significantly upgraded before they can be considered for high-capacity telecommunication systems. Here, we report a new HCF with conjoined-tubes in the cladding and a negative-curvature core shape. It exhibits a minimum transmission loss of 2 dB km$^{-1}$ at 1512 nm and a <16 dB km$^{-1}$ bandwidth spanning across the O, E, S, C, L telecom bands (1302–1637 nm). The debut of this conjoined-tube HCF, with combined merits of ultralow loss, broad bandwidth, low bending loss, high mode quality and simple structure heralds a new opportunity to fully unleash the potential of HCF in telecommunication applications.

[1] Beijing Engineering Research Centre of Laser Technology, Institute of Laser Engineering, Beijing University of Technology, 100124 Beijing, China. [2] Laboratory of Optical Physics, Institute of Physics, Chinese Academy of Sciences, 100190 Beijing, China. Correspondence and requests for materials should be addressed to Y.-y.W. (email: dearyingyingwang@hotmail.com)

The global internet traffic is constantly growing at a very high pace, sparking the general fear of a looming "capacity crunch"[1,2] in the future. Silica-based solid-core fibres have limitations in high-capacity data transmission due to intrinsic material imperfections, i.e. Rayleigh scattering loss (RSL), chromatic dispersion and Kerr nonlinearity. While an optical loss level of 0.14 dB km$^{-1}$ remains sustainable in present long-haul transmission systems and chromatic dispersion can be compensated by dispersion mapping, bulk material nonlinearity represents a more fundamental obstacle for capacity scaling, known as the nonlinear Shannon limit[3,4]. The contemporary scenario of space-division multiplexing (SDM)[5] in association with multicore or multimode fibres cannot fully eliminate the capacity crunch concern, which encourages radical fibre innovations to coordinate with the ever-increasing data transmission demand.

Since its invention[6], hollow-core photonic crystal fibres (HC-PCFs) have been considered as a potentially ideal channel for optical communications. The quasi-vacuum (i.e. no bulk core material) optical environment in HC-PCFs allow light to travel at its fastest speed without experiencing nonlinear effects and dispersion. Over the last two decades, HC-PCFs have developed into a versatile platform for a range of interdisciplinary applications in nonlinear optics[7,8], ultrafast optics[9], high power laser[10], biophotonics[11] and quantum optics[12]. However, the future of HC-PCFs in optical communications is still ambiguous due to several barriers, including their relatively high loss, limited bandwidth and their complex fibre structure. The current transmission loss record for hollow-core bandgap fibres (HC-PBGFs), whose light guidance relies on opening of the out-of-plane photonic bandgap underneath the light line[13], is 1.7 dB km$^{-1}$ at 1620 nm within a bandwidth of only 20 nm[14] or a loss level of 5 dB km$^{-1}$ at 1550 nm for a larger 200 nm bandwidth[15]. These marginal attenuations are intrinsically limited by the surface scattering loss (SSL)[16], while the maximum achieved bandwidth is restrained by the upper limit of air-filling fraction[17] and the surface mode anti-crossing[18], indicating that significant improvement is difficult. Even with the state-of-the-art fabrication, the combination of the minimum loss and the maximum bandwidth in a single fibre is yet to be realized.

Hope has recently resurged with the emergence of hollow-core negative-curvature fibres (HC-NCFs)[19], which originate from the Kagome-type broadband HCF in 2002[20]. After the discovery of the hypocycloid-shape in the core-surround [21,22] and the simplified structure in the cladding[23–25], HC-NCFs attracted massive interest in fibre optics community because: (i) SSL[16] and surface mode effects[26], the two main issues limiting performances of HC-PBGFs, can easily be suppressed in these fibres and, (ii) they offer extra degrees of freedom (e.g. core-surround shape) to control light leakage[27]. Typically, the inhibited coupling[8] or anti-resonant reflecting optical waveguide (ARROW)[28,29] guidance features broadband light transmission, sometimes with octave spanning bandwidth[8], and lower spatial overlap of core mode with silica membrane, which results in a SSL level over one order of magnitude lower than in HC-PBGFs[30,31]. Hence, the main efforts in designing and fabricating HC-NCFs are diverted to the reduction of the confinement loss (CL) and the bending loss (BL). Current record is transmission loss of 7.7 dB km$^{-1}$ at 750 nm and BL of 0.03 dB/turn at bend radius of 15 cm, obtained in a single-ring tube structure[32]. This level of transmission loss and BL are still far from telecom standard but some advanced NCF designs[31,33], e.g. adding nested elements into the cladding tubes, optimistically indicate both ultralow CL and BL. However, with the current pressurized fibre drawing technique and with the consideration of fluid dynamics inside the furnace[34], realization of such delicate structures with predicted optical performance brings substantial challenges due to the difficulty of precisely controlling the size and position of each tube element[35,36,37]. How to remarkably bring down CL and BL by introducing minimum complexity in both fibre design and fabrication becomes a crucial issue for the advancement of HC-NCF in optical communication technology.

In this letter, we report the design, fabrication and characterization of a new HCF structure referred to as hollow-core conjoined-tube negative-curvature fibre (HC-CTNCF or CTF for short). The design concept of this CTF is to conjoin twin (or triplet) anti-resonant (AR) tubes in the radial direction to efficiently confine light by multiple interfaces with sufficient consideration of fabrication simplicity. This new CTF combines almost all the qualities required for future optical communication systems and industrial up-scaling: ultralow transmission loss (2 dB km$^{-1}$), wide bandwidth, low BL, single modedness and simple geometry.

## Results

**Fabricated fibre**. Figure 1a shows the fabricated CTF (see Methods). It consists of six untouched conjoined tubes encircling an air core with a diameter of $D = 30.5\ \mu m$, each formed by conjoining two D-shaped air holes. As light attempts to leak out of the core, it successively encounters five different dielectric layers[38] (or six glass: air interfaces), i.e. the negative-curvature glass layer with the thickness of $t_1 = 1.12\ \mu m$, the first layer of D-shaped air holes with the effective area of $S_1 = 270\ \mu m^2$, the flat glass bar with the thickness of $t_2 = 1.06\ \mu m$, the second layer of D-shaped air holes with the effective area of $S_2 = 335\ \mu m^2$, and the positive-curvature glass layer with the thickness of $t_3 = 1.16\ \mu m$ (labelled in yellow in Fig. 1a). The three glass layers, having nearly the same thickness, operate in the second-order ARROW band[28,29] at the wavelength centred around 1450 nm. The two layers of D-shaped air holes also stay in the first-order ARROW band thanks to careful control of sizes[38]. Accurate numerical simulation (the grey curve in Fig. 1c, see Methods) indicates that these five AR layers collectively bring down the CL by 62 dB from 1210 dB m$^{-1}$ for a capillary bore (with the same core diameter) to 0.7 dB km$^{-1}$ for the CTF at 1512 nm. Experimentally, a cut-back measurement (see Methods) from 330 to 5 m shows a transmission loss of 2 dB km$^{-1}$ at 1512 nm, 3.7 dB km$^{-1}$ at 1550 nm, and 13.4 dB km$^{-1}$ at 1310 nm with the measurement uncertainty <0.1 dB km$^{-1}$ (Supplementary Note 1). The experimentally measured threefold higher losses probably originates from structural non-uniformity in both the transverse and longitudinal dimensions. Additional effort in fabrication is still required to improve structural uniformity in order to achieve sub 1 dB km$^{-1}$ loss level. The bandwidth with optical losses lower than 16 dB km$^{-1}$ extends from 1302 to 1637 nm (covering half of the O band, and full E, S, C, L telecom bands), paving the way towards all the existing wavelength division multiplexing (WDM) transmission patterns (normal, coarse and dense WDM) in one HCF. In this measured spectrum, the loss peak at 1380 nm is due to OH$^-$ absorption which was introduced in the fibre drawing process and could be expelled by dehydroxylation in future trials[39]. Spectral oscillations at band edges, i.e. at 1281, 1295, 1318 and 1620 nm, probably stem from slight inconsistence of the AR conditions for different glass layers which could be optimized by more sophisticated control of the glass membrane thicknesses. Other spectral oscillations distributed over the whole transmission band are the result of Fano resonances[40] introduced by the glass web connections. In this novel NCF design, instead of totally eliminating these connection points[31], we choose to suitably arrange their positions to minimize the negative influences on both the CL and dispersion (Supplementary Note 2) and retain these connections to facilitate adding more AR layers in the cladding area.

**Bending loss**. One fundamental characteristic of an optical fibre is that it allows flexible bending of the light. In a NCF consisting of one ring of ordinary tubes, there exists a tradeoff between the CL and the

BL. Previously reported NCFs mostly pursue low CL by sacrificing BL with the adoption of a large core diameter (30–55 times the working wavelength)[32,41,42], which represents a very small glancing angle at glass:air interfaces. Nevertheless, BL sensitivity is a crucial issue in practical applications and needs to be reduced. Special care was taken in this work to ensure a low BL by designing a core diameter only 20 times the central wavelength, comparable to that of 19 cell HC-PBGF[43], while the ultralow CL is attained by the 5 AR layers in the CTF scheme (Fig. 1a). The increased number of AR layers combined with the reduced core diameter guarantee the simultaneous realization of ultralow CL and low BL. The measured BLs (see Methods) exhibit values as low as 0.7, 1.2 and 5 dB km$^{-1}$

for bending radii $R$ of 10, 7 and 5 cm at 1512 nm, respectively (Fig. 2). The simulated results (the black curves in Fig. 2, see Methods) follow a similar trend. For $R = 5$ cm, a loss peak at 1350 nm is predicted, coinciding with the critical bend-radius formula[44]. However, this peak was not experimentally observed probably due to the indeterministic bending orientations and the small variations of the bending radius along the total 390 loops. Under tighter bending (e.g. $R < 5$ cm), we found the fibre's BL will be notably dependent on the bending direction[45] which requires careful fibre handling. Since a bending radius of 7 cm is acceptable for most fibre applications, the BL-insensitive CTF represents a remarkable step toward practical applications of NCF technology.

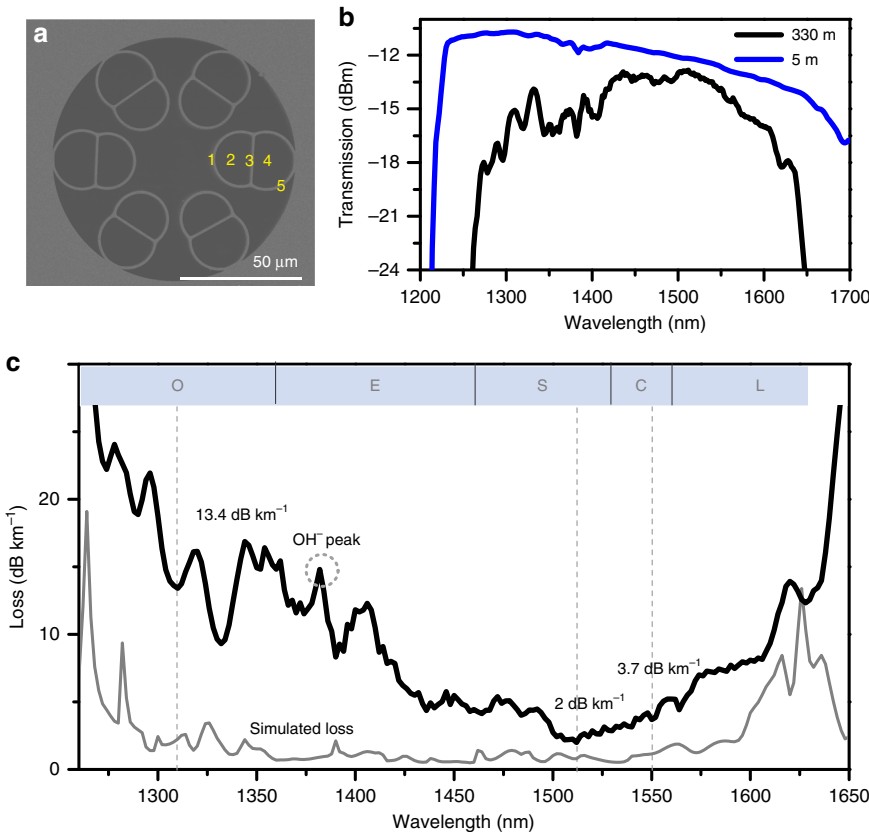

**Fig. 1** Demonstration of an ultralow-loss broadband CTF. **a** Scanning electron microscope (SEM) image of the fabricated CTF with outer diameter of 295 μm. The yellow numbers label the five dielectric layers in the cladding. **b** Measured transmission spectra of the 330 m (black) and 5 m (blue) long CTF under the same launching condition. **c** Measured cut-back loss (black) and simulated loss (grey, including both CL and SSL). The spectral extension of the optical telecommunication O, E, S, C and L bands are also shown for comparison

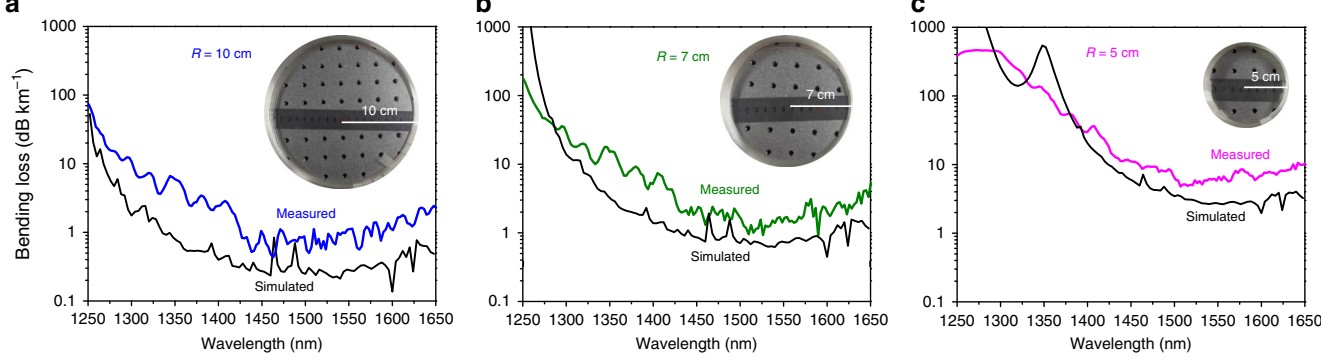

**Fig. 2** Bending loss of the CTF. Measured and simulated bending loss spectra for **a** the bending radius $R = 10$ cm; **b** $R = 7$ cm and **c** $R = 5$ cm. Inset: Photo images of the fibre under test with the loop numbers of 200, 270 and 390, respectively

**Mode quality**. The modal contents of the CTF have been characterized using spectral and spatial (S²) imaging techniques[46] in the wavelength range 1510–1515 nm. For a fibre length $L = 5$ m and a bending radius $R = 48$ cm (nearly straight), the higher order modes (HOM) with group delays of 3.6, 9.2 and 10.8 ps m$^{-1}$ can be discriminated with the multipath interference (MPI) values of $-23$, $-26.5$ and $-21$ dB, corresponding to LP$_{11}$, LP$_{21}$ and LP$_{02}$ modes, respectively (Fig. 3a). When the fibre is bent with a smaller radius, these HOM peaks are suppressed. At $R = 10$ cm, only the peak of LP$_{02}$ mode is visible with the MPI value of $-25$ dB. The

HOM contents could be fully eliminated by increasing the fibre length to $L = 15$ m as shown in Fig. 3b, where no HOM is discernible for both the central launching condition and a deliberately 10 and 15 μm lateral offset launching condition, representing a HOM MPI value >27 dB (limited by the noise floor of the supercontinuum source). Simulation shows CL values of 0.7, 556, 58 and 150 dB km$^{-1}$ for LP$_{01}$, LP$_{11}$, LP$_{21}$, and LP$_{02}$ modes at 1512 nm, respectively (in a straight state), with the lowest HOM extinction ratio of 79 (~19 dB) for the LP$_{21}$ mode. In HC-PBGFs, intentionally designed shunt cores[47] are needed for single modedness. In our CTF, the two D-shaped air holes, having the similar effective indices with the circular tubes of diameter $d = 17.5$ and 20 μm (hence equivalent $d/D$ of 0.57 and 0.66, see Supplementary Note 3), naturally act as the shunt elements by resonantly outcoupling the core HOMs[48]. The CTF therefore could be regarded as a quasi-single-mode fibre for the length >15 m. With more subtle adjustment of the position of the glass bar inside conjoined tube, few mode guidance might be achieved in CTF, which would make them compactible with SDM transmission systems.

## Discussion

According to the multi-layered model[38], in an annular/Bragg HCF, adding one glass layer gives rise to two extra glass:air interfaces. In optimal condition, this could drastically decrease the CL by 22.4 dB (Supplementary Note 4). In a NCF, additional influence from the shape of the glass wall should also be taken into account. In Fig. 4, simulation results indicate that, in comparison with the single-ring NCF, the added glass bar inside the conjoined tube reduces the CL by ~20 dB down to sub 1 dB km$^{-1}$. More surprisingly, a two-bar version CTF with five conjoined tubes and the same core diameter could further decrease the CL to below 0.1 dB km$^{-1}$. From a fabrication perspective, such a 2-bar CTF structure is much more feasible than previously proposed ultralow-loss NCF designs, e.g. nested tube structures[31,33], where the Fano resonances caused by the glass connection points are deliberately avoided but the mechanical stability of the whole structure is less optimized. Our CTF architecture offers another probably more practical route. The simulated CL of the 2-bar CTF (the purple curve in Fig. 4) in the

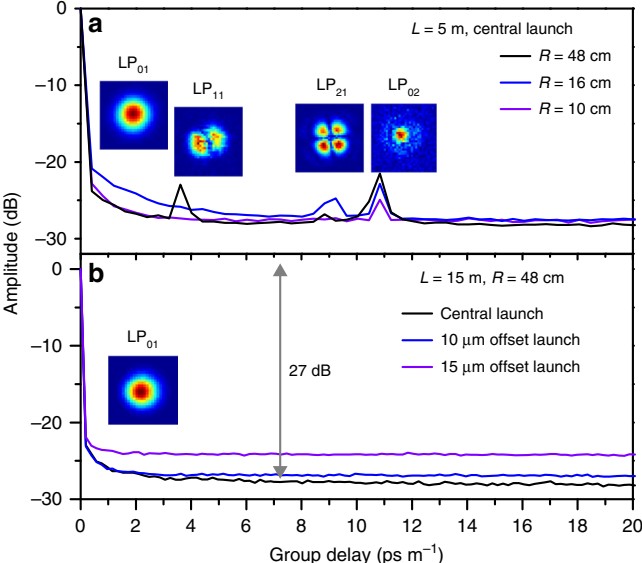

**Fig. 3** S² analysis of the CTF. The typical multipath interference (MPI) vs. differential group delay (DGD) curve for **a** $L = 5$ m fibre with the bending radius of $R = 48$, 16 and 10 cm in central launch condition; **b** $L = 15$ m fibre with $R = 48$ cm in central, 10 μm offset and 15 μm offset launch conditions The insets show the reconstructed mode profiles

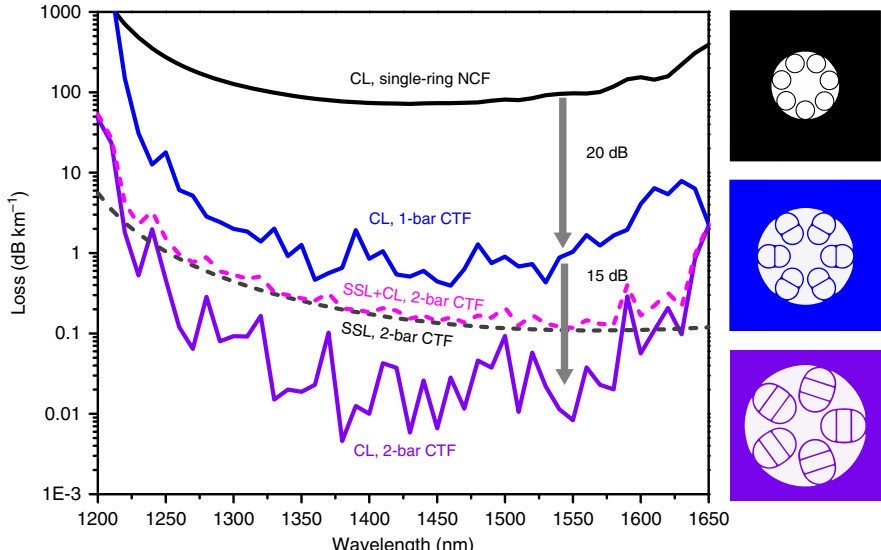

**Fig. 4** Comparison of simulated loss spectra of several designs. Schematic illustrations of the single-ring NCF, 1-bar CTF and 2-bar CTF are shown in the right column in black, blue and purple, respectively. They share the same inscribed core diameter of 30.5 μm. The physical dimensions for each dielectric layers from inward to outward are, $t = 1.12$ μm, $d = 17.5$ μm (same effective index with $S_1$ in CTF, see Supplementary Note 3) for the single-ring NCF; $t_1 = 1.12$ μm, $S_1 = 270$ μm², $t_2 = 1.06$ μm, $S_2 = 335$ μm² for 1-bar CTF; $t_1 = 1.12$ μm, $S_1 = 381$ μm², $t_2 = 1.06$ μm, $S_2 = 564$ μm², $t_3 = 1.06$ μm, $S_3 = 381$ μm² for 2-bar CTF. The CLs are plotted in solid curves. The SSL and the total loss are plotted in dashed curves. Average reductions of 20 dB from the single-ring NCF (~70 dB km$^{-1}$) to 1-bar CTF (~0.7 dB km$^{-1}$) and 15 dB from 1-bar CTF to 2-bar CTF (0.02 dB km$^{-1}$) in CL are estimated from simulation

**Table 1 Optical performance comparison of PBGF, single-ring NCF and CTF**

| Optical performance | PBGF | Single-ring NCF | CTF |
|---|---|---|---|
| Core diameter/wavelength ratio ($D/\lambda$) | 17[43,47] | 55[32] 38[42] | 20 |
| Minimum achieved loss | 1.7 dB km$^{-1}$ @ 1565 nm[14] | 7.7 dB km$^{-1}$ @ 750 nm, $D/\lambda = 55$[32] 20 dB km$^{-1}$ @ 1000 nm, $D/\lambda = 38$[42] | 2 dB km$^{-1}$ @1512 nm |
| Predicted minimum loss | 1.2 dB km$^{-1}$[16] | <1 dB km$^{-1}$ @ $D/\lambda = 55$[32] 10 dB km$^{-1}$ @ $D/\lambda = 38$[39] 70 dB km$^{-1}$ @ $D/\lambda = 20$ (Fig. 4) | 0.11 dB km$^{-1}$ |
| Maximum bandwidth | 200 nm[15] @ < 5 dB km$^{-1}$ | >1000 nm @ < 100 dB km$^{-1}$[42] | 335 nm @ < 16 dB km$^{-1}$ |
| Mode spatial overlap | $2 \times 10^{-3}$ [43] | <$10^{-5}$ [32] | $1.5 \times 10^{-4}$ |
| Dispersion | Tens of ps nm$^{-1}$km$^{-1}$ [15] | <2.5 ps nm$^{-1}$km$^{-1}$ [42] | <6 ps nm$^{-1}$ km$^{-1}$ |
| Group velocity (group index) | 0.29 m ns$^{-1}$ (1.003) [43] | Not reported | 0.2996 m ns$^{-1}$ (1.00067) (calculated) |
| Bending loss | Insensitive to 15–20 mm radii[15] | 0.03 dB/turn (~30 dB km$^{-1}$) for $R = 15$ cm @ 750 nm[32] | <1 dB km$^{-1}$ for $R = 10$ cm @ 1512 nm |
| Mode purity | ~dB m$^{-1}$ loss for HOM @ $R <$ 4.5 cm[47] | >50 dB suppression of HOM for a few tens of metres[42] | >27 dB suppression of HOM for 15 m long fibre |
| PM | ~$10^{-4}$ [42] | Solution needed | Solution needed |

wavelength range of 1325–1585 nm exhibits Fano-induced oscillations varying from 0.005 to 0.09 dB km$^{-1}$, well below the SSL level (0.11–0.24 dB km$^{-1}$). The minor influence to the total loss spectrum (the pink curve in Fig. 4) verifies that the CTF has suppressed the detrimental effects of the glass connection points to a sufficient extent. The size/ellipticity of the conjoined tube and the positions of the glass bars could be regarded as new and unexplored engineering paradigm for future advanced NCF.

In Table 1, a comprehensive comparison of CTF, single-ring NCF and PBGF is listed. As shown in this work, the first trial of the 1-bar CTF has already approached the optical loss level of PBGF with a 20-year history. The demonstrated fibre sample is far from the limit of CTF technique and can be further improved. Similar to the single-ring NCF, CTF could further expand the transmission bandwidth as long as thinner and more uniform glass membranes are realized. Other advantages of CTF over PBGF include lower modal spatial overlap with glass (thus higher laser damage threshold), lower dispersion, and higher group velocity, likening light propagation in vacuum. Although the BL of the CTF is still higher than the PBGF, it is already small enough for industrial application. The remaining issue lies on realization of polarization-maintaining fibres. Further investigation is needed but hybrid transmission bands[49] could represent a potential solution.

To summarize, we believe the CTF demonstrated in this work, with a minimum loss of 2 dB km$^{-1}$, transmission window spanning across O-L telecom bands, low bending sensitivity, high mode purity and simple structure, represents a new milestone towards the long-term goal of ultralow-loss HCF. Although the first trial of CTF has not achieved better optical losses than the more mature PBGF, it has, however, proved that neither CL, BL nor SSL are obstacle for further development of HCF. This makes HCF a strong contender to solid-core silica fibre for high-capacity data communication, where loss and nonlinear tolerance are of equal importance for enhancing spectral efficiency via higher order modulation formats. The next targets would be to upscale the fibre length to multi-kilometre and to optimize the loss level to sub −1 dB km$^{-1}$, which both look viable in CTF and its variants.

## Methods

**Fibre fabrication**. To make conjoined-tubes, a custom-made high purity thin slab is inserted into the centre of a thin glass tube and is drawn into capillaries. The latter is then drawn into fibres using the conventional two-stage stack-and-draw technique.

**Simulation of CL**. The CL is numerically simulated using a finite-element mode solver (COMSOL Multiphysics), with optimized mesh size and perfectly matched layer[50]. The geometrical parameters were determined from the SEM image of the fibre with small adjustment within the range of uncertainties. The SSL is calculated following the equations in ref. [31].

**Transmission loss measurement**. The loss spectrum is obtained by cut-back measurement from 330 to 5 m. A supercontinuum (SC) source is butt-coupled to the 330 m long CTF (inside a fibre splicer), which is looped on the drum during the measurement with $R = 16$ cm. The output end of the CTF is connected to an optical spectral analyzer (OSA) using a magnetic clamp bare fibre adaptor (OZ Optics). Multiple cleaves at the output end show little variation in the recorded spectra for both the 330 and 5 m long fibre samples. More details about the analysis of measurement error can be found in Supplementary Note 1.

**BL measurement**. The CTF's bending loss is measured by comparing the transmission spectrum of a bent fibre with that of a quasi-straight fibre (with the bending radius $R = 50$ cm). The supercontinuum source is butt-coupled to a 130 m CTF, which is bent to $R = 10$, 7 and 5 cm with 200, 270 and 390 loops sequentially (shown in the insets of Fig. 2). The bending loss spectra are demonstrated in Fig. 2.

**BL simulation**. BL is calculated by substracting the loss of a bent fibre from that of a straight fibre. The conformal mapping technique is implemented for the simulation of a bent fibre[51]. Two bending orientation directions, $\theta = 0°$ and $30°$ ($\theta$ as defined in ref. [41]), are taken into account and the two results are averaged.

**S² measurement**. The CTF's S² measurement is performed by launching the SC light into a 5 m (15 m) CTF with different bending radii and the launching conditions are described in the caption of Fig. 3. At the output cross-sectional plane, a SMF28 fibre records the spectra at different positions. The spectra were acquired by an OSA with the resolution of 0.05 nm. The near filed image was also monitored by a camera.

**Data availability**. The data that support the findings of this study are available from the corresponding author upon reasonable request.

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

## Acknowledgements

This work was funded by The National Research and Development Program of China (No. 2017YFB0405200), National Natural Science Foundation of China (NSFC) (No. 61675011, 61527822, 61575218) and Beijing Nova Program (No. Z181100006218097). We would like to thank Dr. Yong Bao for assistance in fibre fabrication and Dr. Kenny Hey Tow for polishing English language.

## Author contributions

S.-f.G. performed the fibre draw, characterization and some simulation. Y.-y.W. and W.D. conceived the project of ultralow-loss NCF and provided the design strategy. S.-f.G. and Y.-y.W. designed the fibre structure. W.D. provided the modelling method. Y.-y.W. and W.D. wrote the manuscript. D.-l.J. performed the S$^2$ measurement. S.G. and X.Z. assisted in fibre fabrication. P.W. supervised the whole project.

## Additional information

**Competing interests:** The authors declare no competing interests.

