## [Peer Review File · Nature Communications]

Reviewers' comments:

Reviewer #1 (Remarks to the Author):

This work reported a significant breakthrough in hollow-core fibers using an innovative conjoined-tube negative curvature design to reduce confinement loss. The reported loss is a record low for negative curvature fibers and is very close to the record loss for photonic bandgap fibers, reported well over a decade ago. The most interesting aspect of the work is the possibility that negative curvature fiber can potentially overcome the limiting surface scattering loss, leading to ultra-low loss fibers. The authors carefully characterized loss, bend loss and mode contents backed up by simulations. The manuscript is well written and referenced relevant previous works. The reported results do not meet the stringent requirement of loss and bend loss for telecommunications but may be useful for a range of other applications such in high-peak power pulse delivery and gas spectroscopy. I recommend the work for publications with some minor revisions.

The authors indicated "Different with the PBG guidance, the inhibited coupling or antiresonant reflecting optical waveguide (ARROW) mechanism allows broader transmission bandwidth...". I do not think ARROW necessarily allow broader transmission bandwidth than PBG. The bandwidth of PBG is mostly limited by modes in the connecting webs and core boundary, which can potentially be minimized by appropriate design.

Reviewer #2 (Remarks to the Author):

The manuscript "Hollow-core conjoined-tube negative-curvature fibre with ultralow loss" reports a HCF with conjoined-tubes in the cladding and a negative curvature in core.

1) The authors show a low loss of 2 dB/km at 1512 nm for their fabricated fibre. This loss value does lower than 7.7 dB/km reported in Ref. 25 for the antiresonant fibre.

2) The main idea in this paper is the use of conjoined tube. The authors show the numerical results that the glass bar inside the conjoined tube reduces the CL by ~ 20 dB. The authors also fabricated this type of fibre with a low loss. However, in the reviewer's point of view, this idea and physics in conjoined tube are not convincing compared to the nested antiresonant fibre suggested in Refs. 24 and 26. The authors use straight bar. However, it is well known that negative curvature in the antiresonant fibre plays a very important role and the straight bar will not be more effective compared with nested tubes suggested in Refs. 24 and 26.

Also, a similar idea to add additional glass layers inside the tubes has already been published in the following paper.

Md Imran Hasan, Nail Akhmediev, and Wonkeun Chang, "Positive and negative curvatures nested in an antiresonant hollow-core fibre," Opt. Lett. 42, 703-706 (2017)

3) At the end the chapter 4, the authors argued that their fibre has wide bandwidth, single modedness and simple geometry.

In terms of bandwidth, the authors show a bandwidth of 1300~1630 nm (<16 dB/km), which corresponds to a bandwidth of $\sim 23\%$ centered at 1450 nm. The figure 5C in Ref. 25 shows a bandwidth of 670~870 nm (<16 dB/km), which corresponds to a bandwidth of $\sim 28\%$ centered at 770 nm. Here, the percentage is used because one can decrease or increase the tube thickness to shift the bandwidth.

For single modedness, it turns out that the authors used bent fibre to show the single mode transmission. Actually, if one uses bent fibre, any antiresonant fibre with a ring of tubes shows some kind of single modedness. The reason is that the higher-order modes will always couple to the cladding mode in bent fibre in certain degree. If the authors want to claim single modedness, it's better to show it in a straight fibre.

In terms of simple geometry, the review believes that the fabrication will be more complicated compared to the simple single-ring NCF showed in Figure 4 of the manuscript and also used in Ref. 25, which has a loss of 7.7 dB/km.

In summary, the reviewer recommends rejection in Nature Communication.

Reviewer #3 (Remarks to the Author):

The manuscript presents a very nice and original idea of improving the design of hollow-core glass fibers for lower transmission loss. There has been a lot of work by other groups speculating about alternative design improvements (e.g. nested capillaries [26]), but this has always been either purely theoretical work, or with the experimentally fabricated structure being far from the intended ideal structure. This work is different in that it not only actually comes up with a realistic design improvement, but also demonstrates the actual fabrication of the structure in an original way (inserting a glass sheet into a tube before drawing it to capillaries). Furthermore, the optical measurements show an amazingly low loss of 2 dB/km at 1512 nm, already the same level as the previous ultralow-loss photonic bandgap fibers, which took years to perfect. Very impressive results!

I do have some questions and suggestions for improvements to the manuscript, which I think should be addressed before it is mature for publication.

1) Minor: the abstract states "<16 dB/km bandwidth covering the O, E, S, C, L telecom bands". It would be nice for the typical reader if the authors here include the minimum and maximum wavelength in nanometers, since not all are familiar with the definitions of the telecom bands.

2) Bottom left paragraph on p. 2:

"with the emergence of hollow core negative-curvature fibre (HC-NCF) [18], which originates from the Kagome type broadband HCF in 2002 [19]. After the discovery of the hypocycloid core shape in 2010 [20,21]..."

This description does not place the work correctly in the context of previous work, and it also is not sufficiently historically correct. It is true that the single-ring NCF can be said to somehow originate from the Kagomé structure, but this is because it was first realized that only the innermost ring of the kagomé cladding really plays a role (S. Février et al., "Understanding origin of loss in large pitch hollow-core photonic crystal fibers and their design simplification," Opt. Express 18, 5142–5150 (2010)). This realization was independent from and did not require the emergence of the hypocycloid shape kagomé, which still had several (mostly unnecessary) rings in the cladding [20, 21]. The negative core-curvature arises naturally when trying to simplify the original kagomé into a single-ring structure, as was done first by the Dianov group: A. D. Pryamikov, et al., "Demonstration of a waveguide regime for a silica hollow - core microstructured optical fiber with a negative curvature of the core boundary in the spectral region > 3.5 μm ," Opt. Express 19, 1441-1448 (2011). It is surprising that this paper is not referenced at all, given its key influence on the design adopted here.

3) The authors refer to Ref. [22] regarding the ARROW mechanism. This is a common mistake, but it would be much more relevant to instead cite the earlier and more foundational work in J.-L.

Archambault et al., "Loss calculations for antiresonant waveguides," *J. Light. Technol.* 11, 416–423 (1993). This work also derives several key equations for the loss of multilayered structures, closely related to the later results in Refs. [S4-S5].

4) In the Results section, it is described that the authors measure a transmission loss of 2 dB/km at 1512 nm when cutting back a fiber from 330 m to 5 m. This means that the difference in measured power at 1512 nm was only $2 \text{ dB/km} \times 0.325 \text{ km} = 0.65 \text{ dB}$, which is less than 0.9% change. What is the uncertainty of this measurement? As pointed out by Prof. Walter Lewin, "Any measurement that you make without the knowledge of its uncertainty is completely meaningless." It is mentioned in the Methods section that "Multiple cleaves at the output end show little variation in spectra for both 330 m and 5 m long fibre respectively", but precisely how much was this "little variation"? If it was on the order of $\sim 1\%$, which could normally be considered "little", it would actually be quite significant relative to the very small change in power at 1512 nm!

And even if the variation between cleaves would be much less than 1%, one should still consider that there are other contributions to measurement uncertainty from e.g. instability in the light source, drift in the coupling from light source to fiber, and from fiber to OSA, etc., so I think it might be overly optimistic to assume that the sum of all these contributions to measurement error is less than 0.9%.

I therefore think it would be highly relevant if the authors include a thoroughly considered error estimate of the measured loss. Otherwise it is difficult to determine how much one can really rely on this 2 dB/km value, originating from a $<0.9\%$ change in measured power.

The authors may of course also consider using a technique such as the multiple cut-back technique (M. Frosz et al., "Five-ring hollow-core photonic crystal fiber with 1.8 dB/km loss", *Opt. Lett.* 38 (13), 2215-2217 (2013)), which directly gives an estimate of the measurement uncertainty.

5) Bottom of right column p. 2: "The usable bandwidth with the loss lower than 16 dB/km", where does the definition of usable bandwidth having less than 16 dB/km loss come from? If it is some industry standard, please provide reference. If it is a more arbitrary limit, please rephrase to make this clearer.

6) In the left column above Fig. 1a: "the loss peaks from 1338 nm to 1421 nm originate from OH-absorption which was introduced in the fibre drawing process and could be expelled by dehydroxylation in future trials." I understand why the authors make the suggestion of the peaks originating from OH-absorption, but I think this cannot be the case. Because there are *multiple* observed distinct peaks in the mentioned region of 1338-1421 nm (Fig. 1c), they look more to be caused by e.g. resonances in the glass web connections. OH-absorption does not cause multiple peaks like these in this wavelength region, but only one smooth distinct peak, see e.g. Figs. 3+4 in O. Humbach et al., *Journal of Non-Crystalline Solids* 203 19-26 (1996), and Fig. 3 in M. H. Frosz et al., "Reducing losses in solid-core photonic crystal fibers using chlorine dehydration", *Optical Materials Express* 6 (9), p. 2975-2983 (2016). The latter paper would maybe also be relevant to cite regarding the authors' mention of later using dehydroxylation to reduce OH-related loss, since this seems to be the only published work on such a procedure for PCFs.

7) The fiber structure shown in Fig. 1a is very nice and impressive! Does the microstructure stay oriented like this all along the fiber? I was wondering if maybe the individual tubes may twist slightly along the fiber, so that the flat glass bars are not always "facing" the core perpendicularly? Did the authors maybe observe this in SEMs of samples taken along the fiber length? Such a longitudinal imperfection could of course provide another explanation for the discrepancy between simulation and experiment?

Also, the work only shows results for one fabricated fiber, so it is natural to ask: how reproducible is the fabrication, and have you made other fibers with similar loss performance?

8) Figure 2: why not put experimental and simulation data together in the same plot for easier comparison? There is no reason to split it up in two figures, and it only makes comparison more difficult.

It is interesting that the authors measure lower bend-loss than expected from the simulations, usually it is the other way around. Do you have any plausible explanations for this? I think there should also be included simulation results when the bend-direction is the same as shown in the inset of Fig. 2b, but where the whole structure is rotated by 30°. It is important to see if this causes higher or lower losses compared to the currently chosen orientation in the simulation, because it might help explain why the experimentally measured losses are lower than the simulated losses.

What is the physical reason for the clear loss-peak in the simulation at 1350 nm for $R=5$ cm? Also, for a simple analytical expression of the critical bend-radius in these types of fibers, please consider if the analysis found in M. H. Frosz et al., "Analytical formulation for the bend loss in single-ring hollow-core photonic crystal fibers", *Photonics Research* 5 (2), 88-91 (2017) might be helpful. The equations found there may also help you explain why the LP21 mode is more suppressed at $R=10$ cm than the LP02 mode! (This fact is not explained by the authors and should be considered a non-trivial observation, since the measured loss of the LP21 mode is lower (58 dB/km) than that of the LP02 mode (150 dB/km) in the straight fiber!)

9) Bottom left of p. 4: "in our CTF the D-shaped air holes in the conjoined tubes naturally act as the shunt elements by resonantly out-coupling the core HOMs [35]". It is indeed interesting that the LP11 mode is not observed at all in in this fiber and apparently effectively suppressed, just as in Ref. [35]. Please calculate and provide in the manuscript the equivalent of the d/D ratio used in Ref. [35] to confirm if it is close to the value of 0.68, which was calculated in Ref. [35] to be optimum for suppression of LP11.

10) Figure 4 compares the losses of a single-ring NCF with a 1-bar CTF, but I am not sure if the structural parameters are chosen so that the comparison is fair? For the single-ring NCF the area S_1 is $367 \mu\text{m}^2$, but for the 1-bar CTF the area S_1 is only $270 \mu\text{m}^2$, so is this a fair comparison? Shouldn't structures used for comparison at least have the S_1 area? Another comparison to check would be to make the structures so that the diameter of a circle inscribed inside the S_1 regions is the same for both structures. I think the manuscript should include at least that of the two possibilities for the single-ring NCF which has the lowest loss (either same S_1 -area or same diameter of a circle inscribed inside the S_1 regions, as for the 1-bar CTF).

11) Table 1 compares only PBGF and the newly suggested CTF. It would improve the value of the paper if the table is expanded with another column for the standard single-ring NCF (you already have most of the calculations from Fig. 4 so there is not much extra work involved). The table would also be more useful if in the row "Modal control" it does not just have a qualitative statement such as "quasi-single mode", but instead presents quantitative information about the HOM-suppression, for example given by a figure-of-merit such as the one applied in Ref. [35].

12) In the description of simulations in the Methods section: "The conformal mapping technique is implemented for the simulation of BL [38]." This is not correct citation technique, one should strive to cite the more original work that provided the analysis and ideas used in subsequent works (such as Ref. [38]). The correct reference for the conformal mapping technique would instead be: M. Heiblum and J. H. Harris, "Analysis of curved optical waveguides by conformal transformation," *IEEE J. Quant. Electron.* 11, 75–83 (1975).

13) According to the Editorial Policy Checklist filled out by the authors, there is supposed to be "A full data availability statement is included in the manuscript", but I do not see any in the manuscript?

Thank you for consideration of our manuscript. We greatly appreciate the efforts of the three reviewers for giving so many constructive suggestions. Please find below our reply to the reviewers point by point.

Reviewer #1 (Remarks to the Author):

This work reported a significant breakthrough in hollow-core fibers using an innovative conjoined-tube negative curvature design to reduce confinement loss. The reported loss is a record low for negative curvature fibers and is very close to the record loss for photonic bandgap fibers, reported well over a decade ago. The most interesting aspect of the work is the possibility that negative curvature fiber can potentially overcome the limiting surface scattering loss, leading to ultra-low loss fibers. The authors carefully characterized loss, bend loss and mode contents backed up by simulations. The manuscript is well written and referenced relevant previous works. The reported results do not meet the stringent requirement of loss and bend loss for telecommunications but may be useful for a range of other applications such in high-peak power pulse delivery and gas spectroscopy. I recommend the work for publications with some minor revisions.

We highly appreciate the reviewer's recommendation.

The authors indicated "Different with the PBG guidance, the inhibited coupling or antiresonant reflecting optical waveguide (ARROW) mechanism allows broader transmission bandwidth...". I do not think ARROW necessarily allow broader transmission bandwidth than PBG. The bandwidth of PBG is mostly limited by modes in the connecting webs and core boundary, which can potentially be minimized by appropriate design.

Complying with this suggestion, we reconsidered the difference between the PBG and ARROW guidance and read the famed paper (OL 29, 2369 (2004)) carefully. We admit that our previous statements are not inappropriate. We rewrote the relative paragraphs and added more related references.

"These marginal attenuations are intrinsically limited by the surface scattering loss (SSL)¹⁶, and the maximum achieved bandwidth is restrained by the upper limit of air-filling fraction¹⁷ and the surface mode anti-crossing¹⁸, indicating that significant improvement is difficult."

"HC-NCF attracted massive attentions in fibre optics community because it not only facilitates the suppression of SSL¹⁶ and surface mode effects²⁶ (the two main issues limiting performances of HC-PBGF) but also exhibits extra degree of freedom (e.g. core-surround shape) to control light leakage. Typically, the inhibited coupling⁸ or anti-resonant reflecting optical waveguide (ARROW)^{27,28} guidance features broadband light transmission (sometimes with octave spanning bandwidth^{8,38}) and lower spatial overlap of core mode with silica membrane (resulting in a SSL level over one order of magnitude lower than HC-PBGF^{29,30})."

We deeply appreciate reviewer 1 for this insightful suggestion!

Reviewer #2 (Remarks to the Author):

The manuscript "Hollow-core conjoined-tube negative-curvature fibre with ultralow loss" reports a HCF with conjoined-tubes in the cladding and a negative curvature in core.

1) The authors show a low loss of 2 dB/km at 1512 nm for their fabricated fibre. This loss value does lower than 7.7 dB/km reported in Ref. 25 for the antiresonant fibre.

We have responded to this point together with point 3).

2) The main idea in this paper is the use of conjoined tube. The authors show the numerical results that the glass bar inside the conjoined tube reduces the CL by ~ 20 dB. The authors also fabricated this type of fibre with a low loss. However, in the reviewer's point of view, this idea and physics in conjoined tube are not convincing compared to the nested antiresonant fibre suggested in Refs. 24 and 26. The authors use straight bar. However, it is well known that negative curvature in the antiresonant fibre plays a very important role and the straight bar will not be more effective compared with nested tubes suggested in Refs. 24 and 26.

Also, a similar idea to add additional glass layers inside the tubes has already been published in the following paper.

Md Imran Hasan, Nail Akhmediev, and Wonkeun Chang, "Positive and negative curvatures nested in an antiresonant hollow-core fibre," Opt. Lett. 42, 703-706 (2017)

Indeed, since 2013, many designs have been proposed with better optical properties than the conjoined-tube structure. But these designs did not give sufficient considerations to the fluid dynamics in the fibre draw. We agree that a negative (or positive) curvature shape is more efficient than straight shape in suppressing light leakage, which has been clarified in our recent model (Ref. 34 in new version). But when one considers practical issues in the fabrication process, in the structural neck-down region, a preform encounters heat, tension, viscosity and pressure before freezing into a slender fibre. A nested negative curvature element may not maintain its original shape. Though active pressurization has been employed, not all the holes can be controlled independently with the state-of-the-art techniques. The nested element is inclined to deform during fabrication. Of course, one could argue that more sophisticated pressurization technique need to be developed and the nested structure could be realized in the end. But to the authors' knowledge and experience, it is a big challenge and no successful demonstration of pressure control to every single hole has been reported yet. So, why not to try other strategies? In another view, if a straight bar inside tube has already reduced the confinement loss by ~ 20 dB, why we bother to pursue a negative curvature bar rather than to add more straight bars? (as shown in Fig. 4 and clearly pointed out in our theoretical model of Ref. 34).

Actions: we have added the reference mentioned by the reviewer. We have also added the reference describing the fluid dynamics inside the furnace (Ref. 33 in the new version) to emphasis the importance of considering practical fibre draw.

3) At the end the chapter 4, the authors argued that their fibre has wide bandwidth, single modedness and simple geometry.

In terms of bandwidth, the authors show a bandwidth of 1300~1630 nm (<16 dB/km) , which

corresponds to a bandwidth of ~23% centered at 1450 nm. The figure 5C in Ref. 25 shows a bandwidth of 670~870 nm (<16dB/km), which corresponds to a bandwidth of ~28% centered at 770 nm. Here, the percentage is used because one can decrease or increase the tube thickness to shift the bandwidth.

For single modedness, it turns out that the authors used bent fibre to show the single mode transmission. Actually, if one uses bent fibre, any antiresonant fibre with a ring of tubes shows some kind of single modedness. The reason is that the higher-order modes will always couple to the cladding mode in bent fibre in certain degree. If the authors want to claim single modedness, it's better to show it in a straight fibre.

In terms of simple geometry, the review believes that the fabrication will be more complicated compared to the simple single-ring NCF showed in Figure 4 of the manuscript and also used in Ref. 25, which has a loss of 7.7 dB/km.

Our complete statement is that our fibre has wide bandwidth, single modedness, simple geometry and *LOW BENDING SENSITIVITY*.

The reviewer compares our fibre with the single ring structure in Ref. 25 (now Ref. 31). This is not a fair comparison and overlooks an important detail that the two fibres don't have the same core size. The well-known loss formula of a capillary (see e.g. OE 23, 1289-1299 (2015)) gives the explicit relationship between fibre loss and core radius r and bend radius R

$$\alpha(r, R, \lambda) \propto k_1 \left(\frac{\lambda^2}{r^3} \right) + k_2 \left(\frac{r^3}{\lambda^2} \frac{1}{R^2} \right)$$

The first term stands for the confinement loss and the second term stands for the additional loss due to a bend (k_1, k_2 are constants). The influences of the core-size-wavelength ratio to the two terms are opposite. In Ref. 31, in order to reduce the first term, the authors choose a r/λ ratio of 27.5 (here we use a r/λ of 10) and measure the fibre loss in an extreme condition, i.e. with the bending radius R of 30 cm. When they measured the loss with $R=15$ cm (i.e. on a standard fibre drum as most researchers do), the bending loss dramatically hoists to 0.03 dB/turn (30 dB/km), even larger than their claimed transmission loss (7.7 dB/km)! With this logic, if we do not manage to decrease k_1 , even lower confinement loss could be realized in even bigger core size fibre. Such a method does not demonstrate better light confinement, whereas the core task of optical fibre is to provide stronger light guidance than capillary. Whether to overcome light diffraction effect forms the essential difference between fibre optics and free-space optics.

On the contrary, we use r/λ ratio of 10 because we want to compare our results with PBGF having a similar core size (as listed in Table 1). Our 2 dB/km loss CTF is measured on a standard fibre drum with $R=16$ cm. Even for R as small as 7 cm, the bending loss is only 1.2 dB/km. This makes our fibre a flexible device for light transmission.

To answer the reviewers questions,

1. In terms of the bandwidth, please note that the two bandwidths are measured in different conditions. The 670-870 nm bandwidth in Ref. 31 is measured under $R=30$ cm. Our 1300-1630 nm bandwidth is measured under $R=16$ cm. The authors in Ref.31 didn't give data about the bandwidth in smaller bending radius. But it is well known that for a NCF, the bandwidth will dramatically shrink under bending (see for example Fig. 9 in OE 24, 7103-7119 (2016)).
2. For singlemodess, we thank the reviewer for the suggestion. We have re-measured the S^2 under the bending radius of 48 cm (perimeter of 3 m) and added the results in the new version. Though for the 5 m long piece, we observed an additional peak for LP_{11} mode, luckily, for the 15

m long fibre, no higher order mode content was observed even at R= 48 cm, indicating that the fibre could be regarded as single mode. For a 5m/15m long fibre, it is practically difficult to measure it in a strictly straight condition. We are lack of such a long optical bench. A bending radius of 48 cm could be regarded as quasi-straight.

3. In terms of simple geometry, this geometry is much simpler than PBGF with a similar r/λ ratio and a similar loss performance. Though single ring NCF has an even simpler structure, it does not belong to ultralow loss fibre.

In summary, the reviewer recommends rejection in Nature Communication.

Reviewer #3 (Remarks to the Author):

The manuscript presents a very nice and original idea of improving the design of hollow-core glass fibers for lower transmission loss. There has been a lot of work by other groups speculating about alternative design improvements (e.g. nested capillaries [26]), but this has always been either purely theoretical work, or with the experimentally fabricated structure being far from the intended ideal structure. This work is different in that it not only actually comes up with a realistic design improvement, but also demonstrates the actual fabrication of the structure in an original way (inserting a glass sheet into a tube before drawing it to capillaries). Furthermore, the optical measurements show an amazingly low loss of 2 dB/km at 1512 nm, already the same level as the previous ultralow-loss photonic bandgap fibers, which took years to perfect. Very impressive results!

We highly appreciate the reviewer's evaluation.

I do have some questions and suggestions for improvements to the manuscript, which I think should addressed before it is mature for publication.

1) Minor: the abstract states "<16 dB/km bandwidth covering the O, E, S, C, L telecom bands". It would be nice for the typical reader if the authors here include the minimum and maximum wavelength in nanometers, since not all are familiar with the definitions of the telecom bands.

Thank you. We have modified it accordingly.

2) Bottom left paragraph on p. 2:

"with the emergence of hollow core negative-curvature fibre (HC-NCF) [18], which originates from the Kagome type broadband HCF in 2002 [19]. After the discovery of the hypocycloid core shape in 2010 [20,21]..."

This description does not place the work correctly in the context of previous work, and it also is not sufficiently historically correct. It is true that the single-ring NCF can be said to somehow originate from the Kagomé structure, but this is because it was first realized that only the inner-most ring of the kagomé cladding really plays a role (S. Février et al., "Understanding origin of loss in large pitch hollow-core photonic crystal fibers and their design simplification," Opt. Express 18, 5142–5150 (2010)). This realization was independent from and did not require the emergence of the hypocycloid shape kagomé, which still had several (mostly unnecessary) rings

in the cladding [20, 21]. The negative core-curvature arises naturally when trying to simplify the original kagomé into a single-ring structure, as was done first by the Dianov group: A. D. Pryamikov, et al., "Demonstration of a waveguide regime for a silica hollow - core microstructured optical fiber with a negative curvature of the core boundary in the spectral region $> 3.5 \mu\text{m}$," Opt. Express 19, 1441-1448 (2011). It is surprising that this paper is not referenced at all, given its key influence on the design adopted here.

Many thanks for such a professional comment. We apologize that Pryamikov's paper (OE 19, 1441–1448 (2011)) was not cited in our previous version. We agree this paper is of vital importance to the development of HC-NCF.

With regard to the history of HC-NCF, at this stage, we find that there does not exist a consistent view and different people see it in different angles. While we agree with the reviewer about the contribution of Pryamikov's work, we also think the hypocycloid core Kagome fibre is important because it is the first experimentally demonstrated NCF with low loss, triggering great interests in exploring the negative curvature core shape. In fact, Pryamikov's paper also mentioned this point --- *"For the first time, an effect of decreasing in the loss level due to a negative curvature of the core boundary was observed for kagome lattice HC MOF"*.

For the effect of cladding structure, S. Fevrier's paper's main contribution is that they simplified the Kagome structure and Pryamikov's paper followed their work with a more elegant design. Although the Kagome fibre contains unnecessary and redundant cladding layers, many researchers deem that it is necessary to add more cladding layers for further reduction of confinement loss in generic HC-NCF structure. So, the essential question is how to add them. Different researchers have different strategies based on their different understandings. Since here we provide a structure with both simplified cladding and negative curvature core, we think it's necessary to highlight the history of both in the introduction. We thereby changed the sentence into *"After the discovery of the hypocycloid-shape in the core-surround^{21,22} and the simplified structure in the cladding²³⁻²⁵,"*. We also added several references.

3) The authors refer to Ref. [22] regarding the ARROW mechanism. This is a common mistake, but it would be much more relevant to instead cite the earlier and more foundational work in J.-L. Archambault et al., "Loss calculations for antiresonant waveguides," J. Light. Technol. 11, 416–423 (1993). This work also derives several key equations for the loss of multilayered structures, closely related to the later results in Refs. [S4-S5].

Many thanks for pointing out this. These two references both have very high impact. The 1993 one is more foundational while the 2002 paper is more related with the fibres. We have added both of them in the new version and in the supplementary.

4) In the Results section, it is described that the authors measure a transmission loss of 2 dB/km at 1512 nm when cutting back a fiber from 330 m to 5 m. This means that the difference in measured power at 1512 nm was only $2 \text{ dB/km} \times 0.325 \text{ km} = 0.65 \text{ dB}$, which is less than 0.9% change. What is the uncertainty of this measurement? As pointed out by Prof. Walter Lewin, "Any measurement that you make without the knowledge of its uncertainty is completely meaningless."

It is mentioned in the Methods section that "Multiple cleaves at the output end show little variation in spectra for both 330 m and 5 m long fibre respectively", but precisely how much was

this “little variation”? If it was on the order of ~1%, which could normally be considered “little”, it would actually be quite significant relative to the very small change in power at 1512 nm!

And even if the variation between cleaves would be much less than 1%, one should still consider that there are other contributions to measurement uncertainty from e.g. instability in the light source, drift in the coupling from light source to fiber, and from fiber to OSA, etc., so I think it might be overly optimistic to assume that the sum of all these contributions to measurement error is less than 0.9%.

I therefore think it would be highly relevant if the authors include a thoroughly considered error estimate of the measured loss. Otherwise it is difficult to determine how much one can really rely on this 2 dB/km value, originating from a <0.9% change in measured power.

The authors may of course also consider using a technique such as the multiple cut-back technique (M. Frosz et al., “Five-ring hollow-core photonic crystal fiber with 1.8 dB/km loss”, Opt. Lett. 38 (13), 2215-2217 (2013)), which directly gives an estimate of the measurement uncertainty.

Actually, 0.65 dB corresponds to a 13.9% decrease. So it's not that difficult to quantify in experiment. We agree with the reviewer that we need to give a thorough consideration of errors. We added a supplementary note to give more details of our measurement and our estimation on the measurement errors. (see supplementary note 1).

We prefer not to use the multiple cut-back technique because it is destructive. Of course, we plan to use this fibre as a whole in some applications, such as optical communication.

5) Bottom of right column p. 2: “The usable bandwidth with the loss lower than 16 dB/km”, where does the definition of usable bandwidth having less than 16 dB/km loss come from? If it is some industry standard, please provide reference. If it is a more arbitrary limit, please rephrase to make this clearer.

We have rephrased it into “the bandwidth with the loss lower than 16 dB/km”

6) In the left column above Fig. 1a: “the loss peaks from 1338 nm to 1421 nm originate from OH-absorption which was introduced in the fibre drawing process and could be expelled by dehydroxylation in future trials.” I understand why the authors make the suggestion of the peaks originating from OH-absorption, but I think this cannot be the case. Because there are *multiple* observed distinct peaks in the mentioned region of 1338-1421 nm (Fig. 1c), they look more to be caused by e.g. resonances in the glass web connections. OH-absorption does not cause multiple peaks like these in this wavelength region, but only one smooth distinct peak, see e.g. Figs. 3+4 in O. Humbach et al., Journal of Non-Crystalline Solids 203 19-26 (1996), and Fig. 3 in M. H. Frosz et al., “Reducing losses in solid-core photonic crystal fibers using chlorine dehydration”, Optical Materials Express 6 (9), p. 2975-2983 (2016). The latter paper would maybe also be relevant to cite regarding the authors' mention of later using dehydroxylation to reduce OH-related loss, since this seems to be the only published work on such a procedure for PCFs.

We thank the reviewer for pointing out this mistake. We checked the loss peaks according to these two papers and we agreed that only the peak at 1380 nm is probably caused by the OH-absorption. We agree that other peaks may stem from Fano resonances. We have modified the text accordingly and added the second reference.

7) The fiber structure shown in Fig. 1a is very nice and impressive! Does the microstructure stay oriented like this all along the fiber? I was wondering if maybe the individual tubes may twist slightly along the fiber, so that the flat glass bars are not always “facing” the core perpendicularly? Did the authors maybe observe this in SEMs of samples taken along the fiber length? Such a longitudinal imperfection could of course provide another explanation for the discrepancy between simulation and experiment?

Also, the work only shows results for one fabricated fiber, so it is natural to ask: how reproducible is the fabrication, and have you made other fibers with similar loss performance?

The fibre structure is indeed very nice! Regarding the twist of the individual tubes, we didn't observe it. Below we show several SEM images taken at the two ends of the 330 m fibre, all stay oriented. When we made the preform, we have carefully adjusted the positions of all the conjoined tubes to make sure that they stay oriented along the 70 cm stack. (In fact, this step is not easy and we had experienced several failures with one of the bars rotated for 10 degrees, resulting in fabricated fibres having a higher loss of ~ 5 dB/km.) Since the 330 m fibre originates from less than 1 cm stack, according to the mass conservation, it should be reasonable to assume that there's no twist inside individual tubes.

However, twisting the structure as a whole along the 330 m fibre may be possible. This may bring some minor effect to the loss, once such twisting is adiabatic. It's not easy to explain all the discrepancies between the simulated and experimental results. Though we have tried our best to profile the real structure in our simulation, they are not perfectly coincident with each other. So the simulation results could only act as a guideline.

With regard to the reproducibility, the key is to make a perfect stack! Actually, we spent quite some time in fabrication and below are several examples of “unperfect” fibres with loss around 5-10 dB/km.

But once the stack has been perfectly arranged, one can obtain quite nice fibres.

[Redacted]

[Figure Redacted]

8) Figure 2: why not put experimental and simulation data together in the same plot for easier comparison? There is no reason to split it up in two figures, and it only makes comparison more difficult.

It is interesting that the authors measure lower bend-loss than expected from the simulations, usually it is the other way around. Do you have any plausible explanations for this? I think there should also be included simulation results when the bend-direction is the same as shown in the inset of Fig. 2b, but where the whole structure is rotated by 30°. It is important to see if this causes higher or lower losses compared to the currently chosen orientation in the simulation, because it might help explain why the experimentally measured losses are lower than the simulated losses.

What is the physical reason for the clear loss-peak in the simulation at 1350 nm for R=5 cm?

Also, for a simple analytical expression of the critical bend-radius in these types of fibers, please consider if the analysis found in M. H. Frosz et al., "Analytical formulation for the bend loss in single-ring hollow-core photonic crystal fibers", *Photonics Research* 5 (2), 88-91 (2017) might be helpful. The equations found there may also help you explain why the LP21 mode is more suppressed at R=10 cm than the LP02 mode! (This fact is not explained by the authors and should be considered a non-trivial observation, since the measured loss of the LP21 mode is lower (58 dB/km) than that of the LP02 mode (150 dB/km) in the straight fiber!)

Firstly, we thank the reviewer for pointing out the discrepancy between the simulated and measured results. We have not thought too much about this discrepancy at the stage of the initial submission. But after pointing out by the reviewer, we analyzed our simulation results of the bending loss (BL) and found a mistake—we didn't subtract the confinement loss of the straight fibre.

In measurement, we used the quasi-straight fibre (R=50 cm) as a benchmark reference. The BL is calculated by

$$BL_{\text{measure}} = [\text{Transmission}(R=10 \text{ cm}) - \text{Transmission}(R=50 \text{ cm})] / L$$

In accordance, in the simulation, BL should be calculated by

$$BL_{\text{simulate}} = \text{Loss}(R=10 \text{ cm}) - \text{Loss}(R=\infty)$$

In our initial simulated results, we forgot to subtract the term of $\text{Loss}(R=\infty)$, resulting in a simulated BL greater than the experimentally measured one.

Below we show the new version of Fig. 2 using the corrected equations. We put the simulated

and measured results in the same diagram for each bending radius (indeed much clearer!). The simulated result is an average of the two orientation directions of 0° and 30° . We have modified the text accordingly.

With regard to the loss peak at $R=5$ cm in the simulation, we used the equation in the PR paper and found that the critical bending radius is 5.2 cm at 1350 nm (using a tube diameter of $20 \mu\text{m}$, see below). This explains this loss peak very well. However, we have not observed this peak in experiment, probably because measurement is the average effect of 390 loops with complicated distributions of bending orientation and bending radius (i.e. $R=5 \text{ cm} \pm 0.3 \text{ cm}$).

With regard to the S2 measurement results of the 5 m fibre (Fig. 3(a)), it is quite difficult to explain the distinction between the LP21 mode and LP02 mode, since they have similar dispersion curves (Bessel coefficients of $u_{21}=5.14$ and $u_{02}=5.52$ are very close). The 58 dB/km and the 150 dB/km are the simulated results. We have not experimentally measured the loss of the higher order modes. We are apt to attribute the phenomena in Fig. 3(a) to the unknown light in-coupling condition.

9) Bottom left of p. 4: “in our CTF the D-shaped air holes in the conjoined tubes naturally act as the shunt elements by resonantly out-coupling the core HOMs [35]”. It is indeed interesting that the LP11 mode is not observed at all in in this fiber and apparently effectively suppressed, just as in Ref. [35]. Please calculate and provide in the manuscript the equivalent of the d/D ratio used in Ref. [35] to confirm if it is close to the value of 0.68, which was calculated in Ref. [35] to be optimum for suppression of LP11.

Actually, after adding the S2 result for $R= 48$ cm (suggested by reviewer 2), the LP11 mode was observed. For the 5 m traces, there might be complicated dynamics between the HOMs since their intensity varies with the bending radius. Luckily, for the 15 m fibre, no HOMs were observed, ensuring our fibre could be regarded as single mode.

We agree with the reviewer that it is necessary to give a d/D ratio. In response to this and the next question, we added a section in the supplementary material (note 3) to calculate the equivalent core and cladding tube diameter. The two conjoined tubes have equivalent n_{eff} with circular tube diameters of $17.5 \mu\text{m}$ and $20 \mu\text{m}$, respectively, resulting in d/D ratio of 0.57 and 0.66 close to the ideal value of 0.68.

10) Figure 4 compares the losses of a single-ring NCF with a 1-bar CTF, but I am not sure if the structural parameters are chosen so that the comparison is fair? For the single-ring NCF the area S_1 is $367 \mu\text{m}^2$, but for the 1-bar CTF the area S_1 is only $270 \mu\text{m}^2$, so is this a fair comparison? Shouldn't structures used for comparison at least have the S_1 area? Another comparison to check would be to make the structures so that the diameter of a circle inscribed inside the S_1 regions is

the same for both structures. I think the manuscript should include at least that of the two possibilities for the single-ring NCF which has the lowest loss (either same S1-area or same diameter of a circle inscribed inside the S1 regions, as for the 1-bar CTF).

We thank the reviewer for pointing out this. Following supplementary note 3, we tried with single ring NCF with tubular sizes of 20 μm and 17.5 μm . The 17.5 μm one has a lower loss value of ~ 70 dB/km (the old version was 120 dB/km). We have changed the trace in Fig.4. We do not modify the 20 dB drop-down arrow because it is only an estimation value and actually depends on where the arrow is. The CL of 1-bar CTF shows variation from 0.4 dB/km to 1.5 dB/km in the central band. If one uses 0.7 dB/km as an average, the drop-down ratio is 20 dB. We have added this information in the figure caption.

11) Table 1 compares only PBGF and the newly suggested CTF. It would improve the value of the paper if the table is expanded with another column for the standard single-ring NCF (you already have most of the calculations from Fig. 4 so there is not much extra work involved).

The table would also be more useful if in the row "Modal control" it does not just have a qualitative statement such as "quasi-single mode", but instead presents quantitative information about the HOM-suppression, for example given by a figure-of-merit such as the one applied in Ref. [35].

We thank the reviewer for this nice suggestion. Initially, we didn't put the values of single ring NCF because they often operate in a larger core size region (D/λ of 40-50), making the comparison not fair. But we do agree it is quite meaningful to add their results. In the modified table, we added one row of D/λ ratio and one column of the single ring NCF with experimental results from Ref. 31 and 38. In the modal control part, we added the HOM suppression ratio for single ring NCF and CTF. For the PBGF, we used the statement from ref. 43. (dB/m loss for HOM @ $R < 4.5$ cm).

12) In the description of simulations in the Methods section: "The conformal mapping technique is implemented for the simulation of BL [38]." This is not correct citation technique, one should strive to cite the more original work that provided the analysis and ideas used in subsequent works (such as Ref. [38]). The correct reference for the conformal mapping technique would instead be: M. Heiblum and J. H. Harris, "Analysis of curved optical waveguides by conformal transformation," IEE J. Quant. Electron. 11, 75–83 (1975).

We have changed the reference accordingly.

13) According to the Editorial Policy Checklist filled out by the authors, there is supposed to be "A full data availability statement is included in the manuscript", but I do not see any in the manuscript?

We have added the statement in the new version. We have also attached the raw data in the new submission.

REVIEWERS' COMMENTS:

Reviewer #2 (Remarks to the Author):

The authors revised the manuscript and improved the quality of the manuscript.

2) In response to the reviewer, the authors agreed that "a negative curvature shape is more efficient than straight shape in suppressing light leakage, ... more sophisticated pressurization technique need to be developed and the nested structure could be realized in the end." The reviewer wants to point out that fibres with nested negative curvature elements have actually been realized and fabricated in several groups already.

W. Belardi, "Design and properties of hollow antiresonant fibers for the visible and near infrared spectral range," *J. Lightw. Technol.*, vol. 33, no. 21, pp. 4497–4503, Nov. 2015.

A. F. Kosolapov, G. K. Alagashev, A. N. Kolyadin, A. D. Pryamikov, A. S. Biryukov, I. A. Bufetov, and E. M. Dianov, "Hollow-core revolver fibre with a double-capillary reflective cladding," *Quantum Electron.* 46(3), 267–270 (2016).

J. E. Antonio-Lopez, S. Habib, A. V. Newkirk, G. Lopez-Galmiche, Z. S. Eznaveh, J. C. Alvarado-Zacarias, O. Bang, M. Bache, A. Schülzgen, and R. A. Correa, "Antiresonant hollow core fiber with seven nested capillaries," *IEEE Photonics Conference (IPC)*, 2016.

Md. Selim Habib, Christos Markos, A. Isa Adamu, J. E. Antonio-Lopez, and Rodrigo Amezcua-Correa, "Visible to Mid-infrared Supercontinuum Generation Using a Gas-filled Hollow-core Fiber," *CLEO 2018*, paper JTh2A.196

In last round of review, the reviewer pointed out that the idea of additional glass layers inside the tubes has already been published in the paper "OL 42 703-706, (2017). The authors just simply added the above reference in the paper, but failed to address the issues that the idea or structure of their fibre in the manuscript has already been published and the impact of this manuscript is low.

3) The authors claim that the comparison between the results from their manuscript and the Ref. 31 is "not a fair comparison". However, there is no direct comparison provided by the authors. Instead, the authors made statement that their results are better using Fig. 9 in OE 24, 7103-7119 (2016), Note that (Fig. 9 in OE 24, 7103-7119 (2016)) only shows the difference between bend radii of 3 cm to 6 cm. The difference between fibres with a bend radius of 3 cm and a bend radius of 6 cm will be much bigger than the difference between fibres with a bend radius of 30 cm (a bandwidth of ~28% from Ref. 31) and a bend radius of 16 cm (a bandwidth of ~23% from this manuscript). In reviewer's opinion, if authors want to make a statement that their fibre is superior, they need to provide direct comparison or rigorous evidence.

The authors claim that they observed a peak for LP₁₁ mode in 5 m long fibre and didn't observe a peak for LP₁₁ mode in 15 m long fibre without providing explanation on their experimental results. Hence, the authors cannot claim single mode for their 5 m long fibre.

Overall, this manuscript demonstrated negative curvature fibre with a low transmission loss. The authors also improved the quality of the manuscript after the revision. However, the idea of additional glass layers inside the tubes has already been published (OL 42 703-706, 2017) and the future development should be nested elements inside the tubes. For the above reasons, the reviewer recommends this work for the publication in a more specialized journal.

Reviewer #3 (Remarks to the Author):

The authors have submitted a very thorough reply to all my questions/comments.

A comment to reviewer #2: the reviewer says

"this idea and physics in conjoined tube are not convincing compared to the nested antiresonant fibre suggested in Refs. 24 and 26. The authors use straight bar. However, it is well known that negative curvature in the antiresonant fibre plays a very important role and the straight bar will not be more effective compared with nested tubes suggested in Refs. 24 and 26".

As the authors also correctly point out, the previous attempts to actually fabricate a good fibre with nested tubes have all failed – the thickness of the nested tubes has always been much larger than that needed for low loss. Even if it would be true that nested tubes are *theoretically* more effective for broadband low loss than the straight bars, one cannot look away from the obvious fact that the authors have actually here fabricated a very impressive working fibre.

Regarding the other comments from reviewer #2, I think the authors have addressed them all very carefully and correctly.

I highly recommend publication of this paper in Nature Communications.

Again we greatly appreciate the efforts of the three reviewers. Please find below our reply to the reviewers point by point.

Reviewer #2 (Remarks to the Author):

The authors revised the manuscript and improved the quality of the manuscript.

2) In response to the reviewer, the authors agreed that "a negative curvature shape is more efficient than straight shape in suppressing light leakage, ... more sophisticated pressurization technique need to be developed and the nested structure could be realized in the end." The reviewer wants to point out that fibres with nested negative curvature elements have actually been realized and fabricated in several groups already.

Actually, Reviewer 3 has helped us in answering this question. "As the authors also correctly point out, the previous attempts to actually fabricate a good fibre with nested tubes have all failed – the thickness of the nested tubes has always been much larger than that needed for low loss. Even if it would be true that nested tubes are *theoretically* more effective for broadband low loss than the straight bars, one cannot look away from the obvious fact that the authors have actually here fabricated a very impressive working fibre."

W. Belardi, "Design and properties of hollow antiresonant fibers for the visible and near infrared spectral range," J. Lightw. Technol., vol. 33, no. 21, pp. 4497–4503, Nov. 2015.

The realized structure in this paper is shown below, where it is obvious that the nested elements are very thick. The realized loss level is 175 dB/km at 480 nm.

A. F. Kosolapov, G. K. Alagashev, A. N. Kolyadin, A. D. Pryamikov, A. S. Biryukov, I. A. Bufetov, and E. M. Dianov, "Hollow-core revolver fibre with a double-capillary reflective cladding," Quantum Electron. 46(3), 267–270 (2016).

Similarly, below is the realized structure in this paper. While their theoretically calculated loss is

4.6×10^{-2} dB/km at 1.09 μm , the realized loss in experiment is 75 dB/km at 1850 nm. The reason lies on the thick glass wall ($\sim 2 \mu\text{m}$) with a big deviation on the wall thickness. Actually, from our fabrication experience, almost any structure with a thick glass wall is not difficult to draw but the resulting fiber always has a poor optical performance because a slight deviation in wall thickness will cause large difference in antiresonant band. To guarantee the optical performance, the fiber usually need to operate in the first or second antiresonant band, meaning that all the glass walls' thickness are around 1 μm or less (for 1550 nm).

J. E. Antonio-Lopez, S. Habib, A. V. Newkirk, G. Lopez-Galmiche, Z. S. Eznaveh, J. C. Alvarado-Zacarias, O. Bang, M. Bache, A. Schülzgen, and R. A. Correa, "Antiresonant hollow core fiber with seven nested capillaries," IEEE Photonics Conference (IPC), 2016.

Below is the fiber structure realized in this paper, with a loss of 2 dB/m at 1100 nm. Actually this image clearly illustrates the usual case of drawing a nested fiber with thin glass wall. The nested elements could not be expanded to have proper functions, resulting in a fiber performance similar to or even worse than single ring fibers. This is why we and many other colleagues in the HCF society search for other solutions, such as the conjoined-tube fibers presented here.

Md. Selim Habib, Christos Markos, A. Isa Adamu, J. E. Antonio-Lopez, and Rodrigo Amezcua-Correa, "Visible to Mid-infrared Supercontinuum Generation Using a Gas-filled Hollow-core Fiber," CLEO 2018, paper JTh2A.196

The fiber structure is similar to the previous one and we didn't find any description on its optical performance.

Another viewpoint about the reviewer's comment is: Based on our recently published theoretical

paper (Wang, Y. Y. & Ding, W., Confinement loss in hollow-core negative curvature fiber: A multi-layered model. *Opt. Express* 25, 33122-33133 (2017)), the advantage of the negative curvature shape of the glass bar, in comparison with a flat one, in reducing CL can be estimated to not greater than 10 dB (see Fig. 4 in that paper), whereas the effect of adding one more glass layer (even not a curved one) plays the dominant role of lowering down the CL. The experimental result in this manuscript combined with the theoretical model actually provides the research community a new route to improve the HCF performance rather than being impeded by the challenging pressure control issue.

Action: To ensure our descriptions are accurate, we modified the sentence “*realization of such complex structures reveals substantial challenges*” into “*realization of such delicate structures with predicted optical performance reveals substantial challenges*” and we added A. F. Kosolapov et al paper as a reference.

In last round of review, the reviewer pointed out that the idea of additional glass layers inside the tubes has already been published in the paper "OL 42 703-706, (2017). The authors just simply added the above reference in the paper, but failed to address the issues that the idea or structure of their fibre in the manuscript has already been published and the impact of this manuscript is low.

From the title of this OL paper “Positive and negative curvatures nested in an antiresonant hollow-core fiber” and its conclusions, “In conclusion, we propose a new type of NC fiber. It has antiresonant tubes nested by an elliptical element that provides active negative and positive curvatures.” one can clearly see that the concept of this paper is no different with other nested structures papers, i.e. to add negative or positive *NESTED* elements inside the structure.

In our manuscript, we have written that “The design concept of this CTF is to conjoin twin (or triplet) anti-resonant (AR) tubes in the radial direction to efficiently confine light by multiple interfaces with sufficient consideration of fabrication simplicity.”

Though there is some similarity in the designed fiber structure, one can clearly see that the design concept of these two works are different. We have completely given up the attempts of keeping the shape of “nested” structures and instead we conjoin twin or triplet tubes whose sizes comply with the AR requirement. Our design philosophy has been elucidated in the theoretical paper (*Opt. Express* 25, 33122-33133 (2017)). As a consequence of this “conjoined tube concept”, which facilitates fibre fabrication, we successfully fabricated such a high performance HCF.

Hence, we think it’s fair to catalog this OL paper as one of the “*advanced fiber design, e.g. adding nested elements into the cladding tubes,*” as already been cited in the manuscript.

With regard to the “impact of this manuscript”, we have emphasized a lot of times that this is an experimental work and should be compared with the other experimental works but not those pure simulation works. We think reviewer 1 and reviewer 3 have given sufficient evaluation on the impact of this work. Experimentally exhibiting the feasibility of a designed structure obviously is much more convincing than theoretical proposal.

3) The authors claim that the comparison between the results from their manuscript and the Ref. 31 is "not a fair comparison". However, there is no direct comparison provided by the authors. Instead, the authors made statement that their results are better using Fig. 9 in OE 24, 7103-7119

(2016), Note that (Fig. 9 in OE 24, 7103-7119 (2016)) only shows the difference between bend radii of 3 cm to 6 cm. The difference between fibres with a bend radius of 3 cm and a bend radius of 6 cm will be much bigger than the difference between fibres with a bend radius of 30 cm (a band width of ~28% from Ref. 31) and a bend radius of 16 cm (a bandwidth of ~23% from this manuscript). In reviewer's opinion, if authors want to make a statement that their fibre is superior, they need to provide direct comparison or rigorous evidence.

We are not sure how to do a "direct comparison" with ref. 31 since ref. 31 shows no data of the transmission bandwidth after bending. Actually we don't think there's any necessity to compare with ref. 31 as the superiority of our fiber is so obvious (as having been highly evaluated by reviewers 1 and 3). The reviewer 2 raised this comparison in the last round of the review, and in response we have already compared with ref. 31 in several aspects. We think we have already give "rigorous evidence" of the superiority of our fibre in the overall performance. The reviewer 3 fairly comments that "*I think the authors have addressed them all very carefully and correctly.*" We are sorry that our last response letter has not convinced reviewer 2, but we really couldn't provide "direct comparison" since there's no paper demonstrating such broad transmission bandwidth even under bending in this loss level before for negative curvature fibre.

The authors claim that they observed a peak for LP11 mode in 5 m long fibre and didn't observe a peak for LP11 mode in 15 m long fibre without providing explanation on their experimental results. Hence, the authors cannot claim single mode for their 5 m long fibre.

We modified our description from "The CTF therefore could be regarded as a quasi-single-mode fibre" into "The CTF therefore could be regarded as a quasi-single-mode fibre for the length greater than 15 m."

Overall, this manuscript demonstrated negative curvature fibre with a low transmission loss. The authors also improved the quality of the manuscript after the revision. However, the idea of additional glass layers inside the tubes has already been published (OL 42 703-706, 2017) and the future development should be nested elements inside the tubes. For the above reasons, the reviewer recommends this work for the publication in a more specialized journal.

We do respect all the efforts having been poured into the nested ARF direction. However, what is the future development is an open question. In this manuscript, we just report our experimental results. We do not think other non-scientific issues like 'future development' should disturb the judgment to our work.

Reviewer #3 (Remarks to the Author):

The authors have submitted a very thorough reply to all my questions/comments.

A comment to reviewer #2: the reviewer says

"this idea and physics in conjoined tube are not convincing compared to the nested antiresonant fibre suggested in Refs. 24 and 26. The authors use straight bar. However, it is well known that negative curvature in the antiresonant fibre plays a very important role and the straight bar will not be more effective compared with nested tubes suggested in Refs. 24 and 26".

As the authors also correctly point out, the previous attempts to actually fabricate a good fibre with nested tubes have all failed – the thickness of the nested tubes has always been much larger than that needed for low loss. Even if it would be true that nested tubes are *theoretically* more effective for broadband low loss than the straight bars, one cannot look away from the obvious fact that the authors have actually here fabricated a very impressive working fibre.

Regarding the other comments from reviewer #2, I think the authors have addressed them all very carefully and correctly.

I highly recommend publication of this paper in Nature Communications.

We highly appreciate reviewer 3's pertinent comments, and his response to reviewer 2 is very helpful.